

# Immediate and cumulative effects of upper-body isometric exercise on the cornea and anterior segment of the human eye

Jesus Vera[1], Beatriz Redondo[1], Rubén Molina[1], Amador García-Ramos[2,3] and Raimundo Jiménez[1]

[1] Department of Optics, Universidad de Granada, Granada, Spain, Spain
[2] Department of Physical Education and Sport, Universidad de Granada, Granada, Granada, Spain
[3] Department of Sports Sciences and Physical Conditioning, Catholic University of Most Holy Concepción, Concepción, Chile, Chile

## ABSTRACT

**Objectives**. The execution of isometric resistance training has demonstrated to cause changes in the ocular physiology. The morphology of the cornea and anterior chamber is of paramount importance in the prevention and management of several ocular diseases, and thus, understating the impact of performing isometric exercise on the eye physiology may allow a better management of these ocular conditions. We aimed to determine the short-term effects of 2-minutes upper-body isometric effort at two different intensities on corneal and anterior eye morphology.

**Methods**. Eighteen healthy young adults performed a 2-minutes isometric biceps-curl exercise against two loads relative to their maximum strength capacity (high-intensity and low-intensity) in a randomized manner. An Oculus Pentacam was used to measure the corneal morphology and anterior chamber parameters in both experimental conditions at baseline, during the isometric effort (after 30, 60, 90 and 120 seconds), and after 30 and 120 seconds of passive recovery.

**Results**. We found that isometric effort causes an increase in pupil size ($P < 0.001$), and a decrease in the iridocorneal angle ($P = 0.005$), anterior chamber volume ($P < 0.001$) and K-flat ($P < 0.001$) during isometric effort, with these effects being more accentuated in high-intensity condition ($P < 0.005$ in all cases). Performing 2-minutes upper-body isometric effort did not alter anterior chamber depth, central corneal thickness, corneal volume, and K-steep ($P > 0.05$ in all cases).

**Conclusions**. Our data exhibit that performing 2-minutes of upper-body isometric exercise modifies several parameters of the corneal morphology and anterior eye biometrics, with these changes being greater for the high-intensity exercise condition. The findings of this study may be of relevance for the prevention and management of corneal ectasias and glaucoma.

Corresponding authors
Jesus Vera, veraj@ugr.es
Beatriz Redondo, beatrizrc@ugr.es

## INTRODUCTION

Physical effort leads to acute changes in ocular physiology, with the effects being dependent on the characteristics of the physical task (*e.g.*, exercise type, exercise intensity, etc.) and the individual performing the task (*e.g.*, fitness level, underlying conditions, etc.) (*Wylęgała, 2016*; *Gale, Wells & Wilson, 2009*; *Zhu, Lai & Choy, 2018*; *Ong et al., 2018*; *Vera et al., 2020*; *Wang et al., 2019*). Isometric efforts, in which force is applied keeping the length of the muscle–tendon unit constant (*Oranchuk et al., 2019*), are present in many daily life tasks (*e.g.*, holding weight) (*Bakke et al., 2007*). This is important since previous investigations have shown that highly demanding isometric efforts cause abrupt alterations in different ocular parameters such as choroidal blood flow (*Kiss, Dallinger & Polak, 2001*; *Tittl, Maar & Polska, 2005*) and intraocular pressure (*Bakke, Hisdal & SO, 2009*; *Vera et al., 2019*).

The Valsalva maneuver is frequently performed when maximal or near maximal isometric efforts are performed in order to facilitate force production by increasing the stabilization of the spine and trunk (*McCartney, 1999*). There is compelling evidence that the corneal morphology, anterior chamber parameters, intraocular pressure, pupil size, choroidal thickness and optic disc topography are sensitive to the execution of the Valsalva maneuver (*Pekel et al., 2014*; *Li et al., 2016*; *Aykan et al., 2010*; *Zhang, Wang & Jonas, 2014*; *Mete et al., 2016*). Regarding anterior eye biometrics, Pekel and colleagues (*Pekel et al., 2014*) found a reduction in keratometric-steep values, central corneal thickness, corneal volume, iridocorneal angle, anterior chamber depth and anterior chamber volume during the execution of the Valsalva maneuver.

Based on the described association between highly demanding isometric efforts and the Valsalva maneuver (*McCartney, 1999*), it is plausible to expect an alteration in the anterior eye biometrics while performing isometric efforts, as has already been demonstrated for the Valsalva maneuver (*Pekel et al., 2014*; *Li et al., 2016*; *Wang et al., 2012*). However, to our knowledge, no study has evaluated this hypothesis. In view of this, we designed a study to assess the impact of performing 2-minutes of upper-body isometric exercise at two different intensities on corneal morphology and anterior chamber parameters. As previously stated, we hypothesized that isometric effort would affect anterior segment morphology (*Pekel et al., 2014*; *Li et al., 2016*; *Wang et al., 2012*), with these changes being accentuated in the more physically demanding condition (*Vera et al., 2019*; *Vera et al., 2019*).

## METHODS

### Participants

An a-priori power analysis, using the GPower 3.1 software (*Faul et al., 2007*) and based on the smallest effect size of interest, was performed to determine the required sample size for this study. For this analysis, we assumed an effect size of 0.25, power of 0.80 and alpha of 0.05. This calculation projected a minimum sample size of 17 participants. At this point, 18 healthy young adults (9 women and 9 men: age = 25.3 ± 5.2 years, body mass = 66.3 ± 12.1 kg, height = 172.4 ± 9.1 cm) were recruited to participate in this investigation. Inclusion criteria were: (i) be free of any ocular disease, as assessed by slit lamp and direct ophthalmoscopy examination, (ii) had no history of refractive surgery or

orthokeratology, and iii) not be currently taking any medication. Participants were asked to refrain from strenuous exercise 48 h preceding each visit to the laboratory, and also to avoid alcohol or caffeine consumption 12 h prior to each testing session. They were also asked to avoid the use of contact lenses for 8 h prior to the main experimental session (second visit). Participants were sport science students from the University of Granada, they were physically active through their standard academic curriculum, which included ∼6 physical activity classes per week (7.9 ± 2.6 h per week). The University of Granada Institutional Review Board (438/CEIH/2017) granted Ethical approval to carry out the study. All participants gave written informed consent prior to the study.

## Anterior eye segment assessment

For the assessment of the eye anterior segment morphology, we used the Pentacam (Oculus Optikgeräte Inc., Wetzlar, Germany). This high-resolution non-contact device uses a rotating Scheimpflug camera, which rotates 360° around the optical axis, and allows to obtain multiple images of the anterior segment of the eye in 2 s, considering the corneal vertex as the reference point (*Shankar et al., 2008*). The acquired data are used by the Pentacam software to construct a 3-dimensional image of the anterior segment, which permits to calculate several anterior segment parameters. For this study, we considered the pupil size (PS; average diameter of the pupil over the duration of the scan), the anterior chamber angle (ACA; the smaller of the 2 angles taken in the horizontal meridian), anterior chamber volume (ACV; volume between the posterior surface of the cornea and the iris and lens over a 12 mm diameter centered on the corneal apex), and the anterior chamber depth (ACD; the distance from the corneal endothelium to the anterior surface of the crystalline). Regarding corneal morphology, we analyzed data of central corneal thickness (CCT; centered at the corneal apex), corneal volume (CV; volume over a diameter of 10 mm centered on the corneal apex), and keratometry readings (K-steep; corneal curvature in the steep central three mm zone, and K-flat; corneal curvature in the flat central three mm zone)

## Upper-body isometric exercise

The isometric biceps curl exercise was always performed bilaterally, in a seated position, with the elbows flexed at 90°. Participants were instructed to maintain a constant breathing pattern (inhaling and exhaling), although an involuntary Valsalva maneuver was expected when leading to muscular failure. The load was applied through two 10-l volume capacity jugs with comfort-grip handles (*Vera et al., 2019*). The heaviest load that each participant could hold for 2 min (13.2 ± 4.1 kg) during the isometric biceps curl exercise was determined in the first testing session. First, participants estimated the load they perceived that they could hold an isometric contraction for a maximum of 2 min. Then, the isometric biceps curl exercise was performed following the technique described above using the 75% of the estimated load. After 30 s of effort, participants were instructed to stop the exercise when they or an experienced researcher perceived that the applied load could be hold for more or less than 2 min. Afterwards, the magnitude of the load was modified in successive sets by consensus between the participant and the researcher until the heaviest load that

each participant could hold for 2 min was identified. Successive sets were separated by 3 min. The main experimental session was performed in a different day and consisted of two sets separated by 10 min. One set was performed holding the heaviest load identified in session 1 (high-intensity condition) and another set holding a load representing half of the heaviest load (low-intensity condition).

## Experimental design and procedure

A repeated measures design (2 intensities ×7 points of measure) was used to test the acute effects of upper-body isometric effort performed at two intensities (high and low) on anterior eye biometrics at seven points of measure (baseline, 30, 60, 90 and 120 s during isometric effort, and 30 and 120 s after isometric effort).

Participants attended the laboratory on two occasions. In the first experimental session, participants were screened for the accomplishment of the inclusion criteria, and we individually determined the heaviest load that each participant could hold for 2 min during the biceps curl exercise (see the "*upper-body isometric exercise*" subsection). In the main experimental session (second visit), upon arrival to the laboratory, participants performed a standardized warm-up consisting of 5 min of jogging and upper-body dynamic stretching exercises. Then, both experimental conditions (high-intensity and low-intensity) were carried out in a randomized order and were separated by 10 min of passive rest. Participants were seated in front of the Pentacam for all the measurements. A total of seven measurements were collected for each experimental condition: one before exercise, four during the 2-minutes isometric exercise (after 30, 60, 90, and 120 s), and two after the isometric exercise (after 30 and 120 s). The isometric biceps curl exercise was always performed bilaterally, in a seated position, and with the elbows flexed at 90°. This position was maintained during the four Pentacam measurements taken during the 2-minutes of upper-body isometric exercise, while the measurements collected before and after exercise were performed in a rested condition (*i.e.,* without performing any isometric effort). All measurements were obtained under constant environmental (∼22 °C and ∼60% humidity) and illumination conditions (∼30 lux).

## Statistical analyses

Before any statistical analysis, the normal distribution of the data (Shapiro–Wilk test) and the homogeneity of variances (Levene's test) were confirmed ($p > 0.05$). To assess the impact of holding weights of different magnitude on corneal morphology and anterior chamber parameters, we performed a repeated measures ANOVA for each dependent variable with the exercise intensity (high, low) and the point of measure (baseline, 30, 60, 90 and 120 s of isometric effort, and 30 and 120 s after isometric effort) as within-participant factors. The level of statistical significance was set at 0.05, and multiple comparisons were corrected with the Holm–Bonferroni procedure. The Cohen's *d* effect size (*d*) and eta squared ($\eta_p^2$) were assessed to determine the magnitude of the differences for T and F tests, respectively. The criteria for interpreting the magnitude of the Cohen's effect size were: trivial (<0.20), small (0.20–0.50), moderate (0.60–1.19), large (1.20–2.00), and extremely large (>2.00) (*Hopkins et al., 2009*).

## RESULTS

Table 1 shows the descriptive values for all the dependent variables assessed at the different points of measure.

### Pupil size

PS exhibited a statistically significant effect for the intensity ($F_{1,17} = 13.05$, $P = 0.002$, $\eta^2_p = 0.43$), point of measure ($F_{6,102} = 6.92$, $P < .001$, $\eta^2_p = 0.29$), and the interaction "intensity $\times$ point of measure" ($F_{6,102} = 4.95$, $P < .001$, $\eta^2_p = 0.23$). After it, we performed a separate one-way ANOVAs for each isometric effort intensity (high and low) with the point of measure as the within-participants factor. For the high-intensity condition, the point of measure reached statistical significance ($F_{6,102} = 10.47$, $P < .001$, $\eta^2_p = 0.38$) due to a lower PS at baseline compared to 30 (corrected $P$-value $= .001$, $d = 1.24$), 60 (corrected $P$-value $= .006$, $d = 1.05$), 90 (corrected $P$-value $= .021$, $d = 0.86$) and 120 (corrected $P$-value $= .020$, $d = 0.89$) seconds during isometric effort. The ANOVA for the low-intensity condition did not show statistical significance for the point of measure ($F_{6,102} = 1.87$, $P = .092$) (Fig. 1).

### Anterior chamber parameters

The analysis of the ACA exhibited a statistically significant effect for the point of measure ($F_{6,102} = 3.35$, $P = .005$, $\eta^2_p = 0.16$), but no differences were obtained for the intensity ($F_{1,17} = 0.70$, $P = .416$) or the interaction "intensity $\times$ point of measure" ($F_{6,102} = 0.61$, $P = .718$). Consequently, we carried out separate analyses for each exercise intensity. Statistical significance differences for the point of measure were showed for the high-intensity condition ($F_{6,102} = 3.11$, $P = .008$, $\eta^2_p = 0.15$), but not for the low-intensity condition ($F_{6,102} = 0.84$, $P = .543$). Post-hoc analyses for the high-intensity condition evidence a narrower ACA compared to baseline after 30 (corrected $P$-value $= .018$, $d = 0.95$), 90 (corrected $P$-value $= .047$, $d = 0.84$) and 120 (corrected $P$-value $= .034$, $d = 0.88$) seconds of isometric exercise (Fig. 2A).

The ACV reached a statistically significant effect for the point of measure ($F_{6,102} = 9.95$, $P < .001$, $\eta^2_p = 0.37$), whereas the intensity ($F_{1,17} = 3.08$, $P = .098$) and the interaction "intensity $\times$ point of measure" ($F_{6,102} = 1.03$, $P = .412$) did not reveal statistical significance. The two separate one-way ANOVAs for each intensity revealed a statistically significant effect of the point of measure for both the high-intensity ($F_{6,102} = 7.83$, $P < .001$, $\eta^2_p = 0.32$) and low-intensity ($F_{6,102} = 4.98$, $P < .001$, $\eta^2_p = 0.23$) conditions. Post-hoc comparisons revealed for the high-intensity condition a reduced ACV compared to baseline after 120 s of isometric effort (corrected $P$-value $= .002$, $d = 1.20$), as well as after 30 s of passive recovery (corrected $P$-value $= .021$, $d = 0.92$). No post-hoc differences were observed in the low-intensity condition (Fig. 2B). Lastly, the analysis of ACD did not show differences for the intensity ($F_{1,17} = 1.86$, $P = .191$), point of measure ($F_{6,102} = 1.56$, $P = .165$), or the interaction "intensity $\times$ point of measure" ($F_{6,102} = 0.60$, $P = .727$).

### Pachymetry

CV did not show statistically significant differences for the intensity ($F_{1,17} = 0.05$, $P = .819$), point of measure ($F_{6,102} = 0.38$, $P = .889$), or the interaction "intensity $\times$ point of

**Table 1  Average ± standard deviation values of the anterior eye segment parameters obtained at the different points of measure during the high-intensity and low-intensity isometric conditions.**

| | | Baseline | Upper-body isometric effort | | | | Recovery | |
|---|---|---|---|---|---|---|---|---|
| | | | 30 sec | 60 sec | 90 sec | 120 sec | 30 sec | 120 sec |
| **Pupil size (mm)**[* # $] | High | 3.27 ± 0.45 | 3.51 ± 0.53 | 3.55 ± 0.54 | 3.51 ± 0.48 | 3.55 ± 0.52 | 3.32 ± 0.49 | 3.24 ± 0.40 |
| | Low | 3.27 ± 0.47 | 3.28 ± 0.41 | 3.24 ± 0.50 | 3.17 ± 0.46 | 3.19 ± 0.43 | 3.13 ± 0.37 | 3.14 ± 0.40 |
| **Anterior chamber angle (degrees)**[#] | High | 41.11 ± 4.40 | 38.83 ± 4.66 | 38.81 ± 4.71 | 38.47 ± 5.11 | 38.51 ± 4.33 | 38.81 ± 4.68 | 39.88 ± 4.62 |
| | Low | 40.61 ± 5.79 | 39.33 ± 4.17 | 39.38 ± 4.39 | 39.17 ± 4.58 | 39.25 ± 4.66 | 39.54 ± 4.49 | 39.22 ± 4.40 |
| **Anterior chamber volume (mm³)** [#] | High | 183.40 ± 29.27 | 183.86 ± 29.81 | 180.59 ± 30.21 | 180.98 ± 31.30 | 177.20 ± 29.67 | 176.57 ± 29.28 | 178.71 ± 28.21 |
| | Low | 184.86 ± 29.02 | 184.11 ± 29.45 | 184.21 ± 28.19 | 181.04 ± 28.23 | 179.62 ± 28.97 | 179.53 ± 29.48 | 179.66 ± 29.27 |
| **Anterior chamber depth (mm)** | High | 3.12 ± 0.26 | 3.10 ± 0.27 | 3.09 ± 0.26 | 3.09 ± 0.25 | 3.08 ± 0.26 | 3.11 ± 0.25 | 3.11 ± 0.25 |
| | Low | 3.12 ± 0.24 | 3.11 ± 0.25 | 3.11 ± 0.25 | 3.12 ± 0.27 | 3.11 ± 0.26 | 3.12 ± 0.26 | 3.11 ± 0.26 |
| **Corneal volume (mm³)** | High | 63.06 ± 2.97 | 63.06 ± 2.85 | 63.18 ± 3.06 | 63.30 ± 3.23 | 63.42 ± 3.11 | 63.08 ± 2.96 | 63.09 ± 2.78 |
| | Low | 63.04 ± 2.91 | 63.09 ± 3.00 | 63.17 ± 2.83 | 63.17 ± 2.84 | 63.18 ± 3.00 | 63.12 ± 2.90 | 63.12 ± 2.88 |
| **Central corneal thickness (μm)** | High | 559.06 ± 28.80 | 561.22 ± 28.00 | 563.72 ± 24.92 | 562.11 ± 28.82 | 562.89 ± 26.50 | 559.06 ± 26.52 | 561.56 ± 28.19 |
| | Low | 560.61 ± 27.35 | 560.06 ± 29.90 | 560.56 ± 27.57 | 560.22 ± 28.29 | 559.78 ± 28.55 | 558.89 ± 29.94 | 561.89 ± 27.28 |
| **K-steep (D)** | High | 43.54 ± 1.91 | 43.55 ± 1.92 | 43.60 ± 2.00 | 43.52 ± 1.95 | 43.57 ± 1.96 | 43.58 ± 1.91 | 43.61 ± 1.96 |
| | Low | 43.53 ± 1.91 | 43.58 ± 1.91 | 43.57 ± 1.95 | 43.55 ± 1.91 | 43.54 ± 1.95 | 43.58 ± 1.97 | 43.60 ± 1.97 |
| **K-flat (D)**[#] | High | 42.76 ± 1.56 | 42.69 ± 1.55 | 42.66 ± 1.55 | 42.62 ± 1.49 | 42.58 ± 1.51 | 42.71 ± 1.51 | 42.71 ± 1.53 |
| | Low | 42.75 ± 1.58 | 42.68 ± 1.52 | 42.67 ± 1.46 | 42.66 ± 1.52 | 42.67 ± 1.47 | 42.71 ± 1.52 | 42.70 ± 1.55 |

**Notes.**
Statistical significance ($p < 0.05$) for the main effects of "intensity" and "point of measure", and the interactive effect of "intensity × point of measure" are depicted by *, #, and $, respectively.

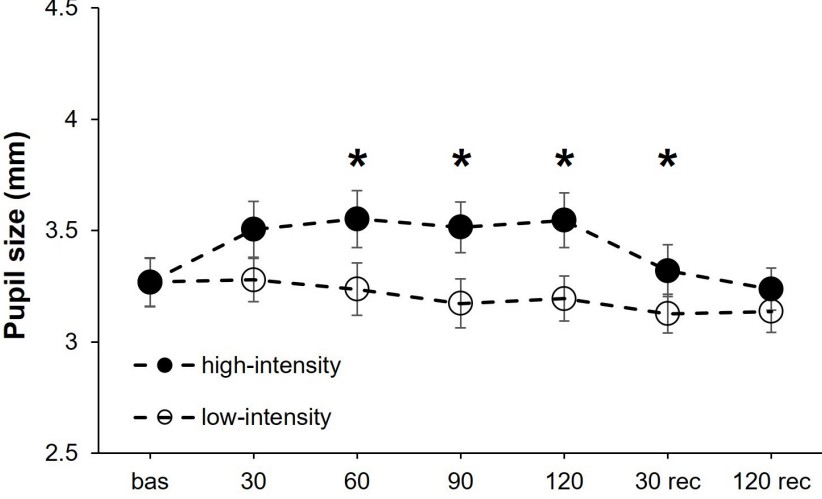

**Figure 1  Pupil size changes at both exercise intensities over time.** Effects of upper-body isometric exercise performed at high and low intensities on pupil size at the different points of measure. The asterisk denotes statistically significant differences between the high-intensity and low-intensity conditions (corrected $p$-value $< 0.05$). The error bars represent the standard error. Bas = baseline; rec = recovery.

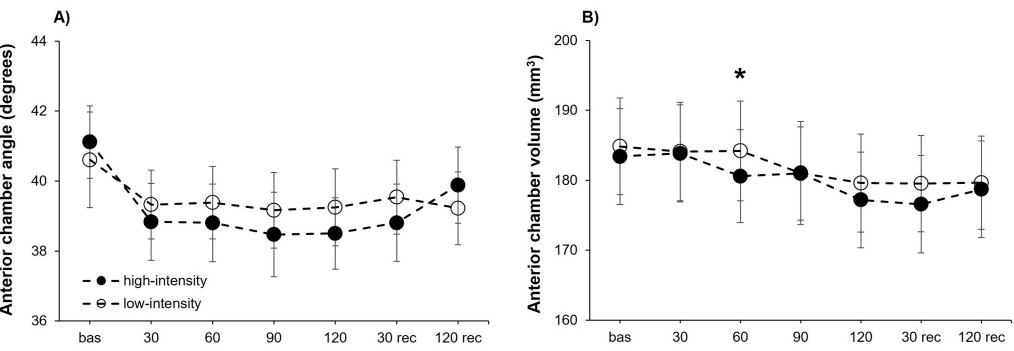

**Figure 2 Anterior chamber angle and volume changes at both exercise intensities over time.** Effects of upper-body isometric exercise performed at high and low intensities on anterior chamber angle (panel A) and anterior chamber volume (panel B) at the different points of measure. The asterisk denotes statistically significant differences between the high-intensity and low-intensity conditions (corrected $p$-value < 0.05). The error bars represent the standard error. Bas = baseline; rec = recovery.

measure" ($F_{6,102} = 0.46$, P = .996). Similarly, the main factors of intensity ($F_{1,17} = 0.73$, $P = 0.406$) and point of measure ($F_{6,102} = 0.90$, P = .497), as well as the interaction "intensity × point of measure" ($F_{6,102} = 0.46$, P = .837) were far from reaching statistical significance for CCT.

### Keratometry readings

For K-flat, a significant effect was found for the point of measure ($F_{6,102} = 5.29$, $P < .001$, $\eta^2_p = 0.24$), whereas no differences were obtained for the intensity ($F_{1,17} = 0.41$, P = .528) or the interaction "intensity × point of measure" ($F_{6,102} = 1.44$, P = .208). Separate unifactorial ANOVAs revealed a statistically significant effect of the point of measure for the high-intensity condition ($F_{6,102} = 5.09$, $P < .001$, $\eta^2_p = 0.23$), but not for the low-intensity condition ($F_{6,102} = 1.88$, P = .092). Post-hoc analyses evidenced in the high-intensity condition a lower K-flat compared to baseline measurements at 90 (corrected $P$-value = .006, $d = 0.88$) and 120 (corrected $P$-value = .001, $d = 1.10$) seconds of isometric effort. Also, the K-flat measured after 120 s of upper-body isometric exercise was lower than the obtained after 30 (corrected $P$-value = .017, $d = 0.81$) and 120 (corrected $P$-value = .029, $d = 0.78$) seconds of passive recovery (Fig. 3A).

The analysis of K-steep did not reach statistically significant differences for the point of measure ($F_{6,102} = 2.06$, P = .069), the intensity ($F_{1,17} = 0.01$, P = .938) and the interaction "intensity × point of measure" ($F_{6,102} = 0.48$, P = .825) (Fig. 3B).

### DISCUSSION

The present study was designed to assess the short-term effects of 2-minutes upper-body isometric effort performed at two intensities on corneal morphology and anterior chamber parameters. Our data evidenced a greater PS for the high-intensity isometric condition, whereas PS remained stable in the low-intensity condition. Regarding anterior chamber parameters, there was a reduction of the iridocorneal angle and ACV during

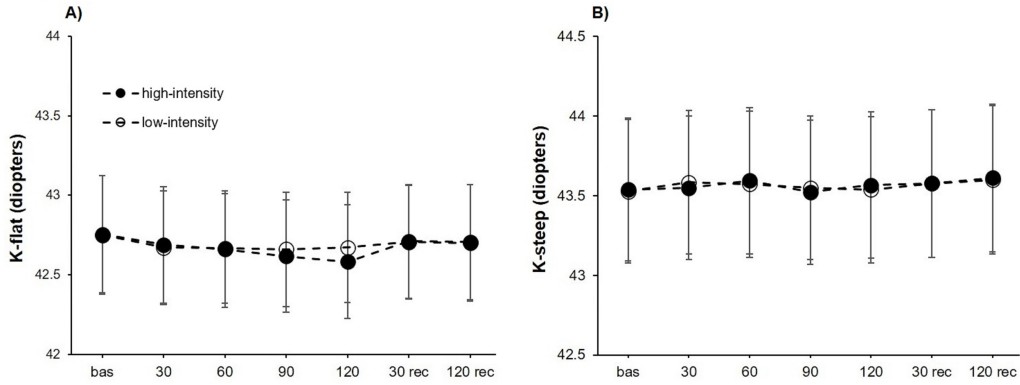

**Figure 3  Keratometry reading changes at both exercise intensities over time.** Effects of upper-body isometric exercise performed at high and low intensities on K-flat (A) and K-steep (B) at the different points of measure. The error bars represent the standard error. Bas = baseline; rec = recovery.

the isometric effort, with these effects being more evident in the high-intensity condition in comparison to the low-intensity condition. Also, a small, but statistically significant, reduction of the K-flat occurred during isometric effort in the high-intensity condition. Taken together, these findings highlight that the anterior eye biometrics are affected by the execution of isometric effort. The outcomes of this study may be of special relevance for the prevention and management of different ocular conditions such as corneal ectasias and glaucoma (*Miglior & Bertuzzi, 2013*; *Brautaset, Nilsson & Miller, 2013*).

A recent study has demonstrated a very high reliability of the Pentacam for measuring different variables (*i.e.,* ACA, ACD and CCT) using similar timestamps as the ones implemented in our study (*Vera et al., 2019*). Therefore, the changes observed in the current study are unlikely to be caused by the multiple measurements, and they should be attributed to our experimental manipulation. Regarding PS, our results agree with *Zénon, Sidibé & Olivier (2014)* who also found a systematic increase in PS during the execution of a handgrip contractions, with the magnitude of the changes being positively associated with the intensity of the physical effort. Zenon and colleagues (*Zénon, Sidibé & Olivier, 2014*) stated that this effect may be caused by the shared neurological mechanisms involved in physical effort and pupil dilation. Also, greater pupil diameters have been observed during the execution of the Valsalva maneuver (*Mete et al., 2016*; *Sun et al., 2020*), and this may partially explain the larger PS observed for the high-intensity condition due to the involuntary execution of the Valsalva maneuver during very physically demanding efforts.

There is scarce evidence about the impact of isometric resistance training on the ocular physiology, with most of these studies being focused on intraocular pressure (*Bakke, Hisdal & SO, 2009*; *Vera, Jiménez & Redondo, 2019*; *Vera et al., 2019*; *Vera et al., 2021*; *Castejon et al., 2010*). Indeed, Vera and colleagues (*Vera et al., 2020*) recently reported that performing the same physical effort used in this study (*i.e.,* 2 min of biceps-curl isometric effort leading to muscular failure) promotes an average intraocular pressure rise of ∼25% (approximately four mmHg). In line with this result, we found an ACA narrowing of

approximately 2.5 degrees during the execution of the high-intensity biceps-curl isometric exercise. Nevertheless, the causal relationship between the IOP rise and ACA narrowing observed during isometric effort cannot be ruled because both parameters were not assessed simultaneously.

Here, we found an effect of isometric effort on the anterior chamber morphology. Specifically, ACA and ACV were reduced during the execution of upper-body isometric exercise, with these changes being more evident for the high-intensity condition. These results are in line with studies assessing the impact of the Valsalva maneuver on anterior eye biometrics (*Pekel et al., 2014*; *Li et al., 2016*; *Mete et al., 2016*; *Wang et al., 2012*; *Sun et al., 2020*). For example, Li and colleagues (*Li et al., 2016*) reported that ACA and ACV decreased sharply when performing the Valsalva maneuver in healthy subjects. Also, a recent study assessing the influence of holding weights on ACA and ACD (*Vera et al., 2019*), found that holding weights corresponding to 20% of participants' weight caused a significant ACA narrowing, whereas ACD was insensitive to this physical effort. These findings agree with the current results, since we found an ACA reduction, but ACD remained stable during the execution of the upper-body isometric exercise.

Lastly, to the best of our knowledge, no previous study has investigated the impact of isometric effort on the corneal morphology. A recent study observed that holding weights corresponding to 10% and 20% of participants' weight did not modify CCT (*Vera et al., 2019*). Regarding the Valsalva maneuver, there are mixed results in the scientific literature, with these differences being possibly attributable to the different techniques used to execute the Valsalva maneuver (*Pekel et al., 2014*; *Mete et al., 2016*; *Mete et al., 2016*; *Duru et al., 2017*). We found a decrease in K-flat during isometric effort. Similarly, there is a lack of studies assessing the changes in corneal morphology associated with physical effort. As previously stated, studies exploring the impact of the Valsalva maneuver on the corneal morphology have reported controversial findings, although the technique used for the Valsalva maneuver also differ between studies (*Pekel et al., 2014*; *Mete et al., 2016*; *Mete et al., 2016*; *Duru et al., 2017*). Nevertheless, and based on the current findings, the impact of isometric exercise on the corneal morphology seems to be modest.

There are claims that multiple daily activities (*e.g.*, eye rubbing, yoga, weightlifting, wearing swimming googles, sleeping position, caffeine intake, etc.) lead to changes in the corneal morphology and anterior eye biometrics, which may have an influence on the incidence and progression of corneal ectasias and glaucoma (*Anderson, 2019*; *Hecht et al., 2017*; *Hashemi et al., 2020*; *Mazharian et al., 2020*; *Redondo et al., 2020*; *Jiménez et al., 2020*). The present study shows that upper-body isometric exercise should be added to this range of activities that alter the anterior eye segment. It should be stated that the clinical relevance and long-term effects of performing physically demanding isometric efforts on the eye physiology require further investigation. However, eye care practitioners should be aware of the relevance of lifestyle habits that involves isometric muscular efforts in order to establish the most pertinent recommendations for the prevention and management of different eye conditions. This study incorporates novel insights into the short-term effects of isometric effort on the corneal and anterior eye morphology, however, our experimental design has several limitations that should be considered when interpreting our findings.

The inclusion of individuals with different eye conditions would allow to better determine the clinical relevance of performing isometric exercise on eye health. Also, changes in the ocular physiology during physical effort have demonstrated to be dependent on exercise' characteristics and individuals' fitness level (*Vera et al., 2020*; *Rüfer et al., 2014*; *Vera et al., 2018*). Thus, further investigations should assess the role of these factors on the corneal and anterior chamber alterations provoked by physical exercise. Mixed results have been obtained for the sex differences on the intraocular pressure changes caused by exercise (*Vera et al., 2019*; *Vera et al., 2020*; *Gene-Morales et al., 2021*; *Pérez-Castilla et al., 2020*), and the possible mediating effect of participants' sex on corneal and anterior segment changes during exercise remains unknown. Future studies are necessary in this regard. The age range of our experimental sample was limited to young adults and the isometric effort only included a specific joint angle. Therefore, the external validity of our findings to older individuals or when the exercise is performed at different joint angles should be addressed in future investigations. Lastly, the possible long-term effects of isometric effort on the cornea and anterior chamber of the eye are unknown, and it is our hope that future studies would explore the incidence of eye conditions on individuals who routinely perform highly demanding physical efforts (*e.g.*, powerlifters or bodybuilders). We can speculate that repetitive changes on corneal and anterior chamber morphology due to isometric effort could lead to long-term consequences, however, follow-up studies are needed to obtain valid conclusions on this matter.

Our data show that performing a 2-minutes upper-body isometric effort causes an increase in PS, and reduces ACA, ACV and K-flat, with these effects being more evident when the exercise is performed at higher intensities. The changes in corneal morphology and anterior chamber parameters rapidly returned to baseline levels after 30 s of passive recovery. Based on the current findings, highly demanding upper-body isometric efforts must be incorporated into the list of activities or factors that can acutely alter corneal morphology and anterior chamber parameters. The inclusion of individuals with ocular conditions dependent on the morphology of the cornea and anterior chamber will allow to better ascertain the clinical relevance of the current findings.

### Funding

The University of Granada ("Plan Propio" program; grant reference: PJIA2020.13) provided financial support in the form of grant funding. The funders had no role in study design, data collection and analysis, decision to publish, or preparation of the manuscript.

### Grant Disclosures

The following grant information was disclosed by the authors:
The University of Granada: PJIA2020.13.

### Competing Interests

Jesús Vera and Amador Garcia-Ramos are Academic Editors for PeerJ.

## Author Contributions

- Jesus Vera conceived and designed the experiments, performed the experiments, analyzed the data, prepared figures and/or tables, authored or reviewed drafts of the paper, and approved the final draft.
- Beatriz Redondo conceived and designed the experiments, performed the experiments, authored or reviewed drafts of the paper, and approved the final draft.
- Rubén Molina performed the experiments, authored or reviewed drafts of the paper, and approved the final draft.
- Amador García-Ramos performed the experiments, analyzed the data, authored or reviewed drafts of the paper, and approved the final draft.
- Raimundo Jiménez conceived and designed the experiments, authored or reviewed drafts of the paper, and approved the final draft.

## Human Ethics

The following information was supplied relating to ethical approvals (i.e., approving body and any reference numbers):

The University of Granada Institutional Review Board (438/CEIH/2017) granted Ethical approval to carry out the study.

## Data Availability

The raw measurements are available in the Supplementary File.

## Supplemental Information

Supplemental information for this article can be found online at http://dx.doi.org/10.7717/peerj.13160#supplemental-information.

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
