# Peer review of "Immediate and cumulative effects of upper-body isometric exercise on the cornea and anterior segment of the human eye"

_PeerJ, doi:10.7717/peerj.13160_

## Round 0.1 · original submission · Minor Revisions

The two expert reviewers have found great merit in your article. They have also raised valuable points needing clarification for you to respond to. In particular, both reviewers have mentioned the relative lack of description of the exercise protocol, and the need to further describe limitations regarding participant age and sex.

Reviewer 1 ·

Basic reporting

No comment

Experimental design

No comment

Validity of the findings

No comment

Additional comments

This is an interesting and robust study aimed at evaluating the corneal and anterior segment of the eye biomechanical adaptations to upper-body isometric efforts. Thank you very much for the opportunity of reviewing this article that greatly adds to the scientific body of knowledge. I congratulate the authors for the work performed. Hereunder, a few minor comments that could help the readers in the understanding of the article and its background are presented:

In the abstract (line 45) the authors point out that the changes are positively associated with the intensity. I believe that with this statement the reader could understand correlations were performed. Consider rewriting it.

A short definition of isometric contraction (e.g., where the muscle-tendon unit remains at a constant length [Oranchuk et al., 2019]) could be inserted in the introduction to help potential readers as this is not a sport-specialized journal.

In the a-priori sample size analysis, was the effect size calculated based on previous studies? on pilot studies? on the smallest effect size of interest?

Please, consider including information on the fitness level of the participants as this measure could influence ocular physiological responses. This information could be objective (e.g., the maximum isometric force produced, kilograms used for the 2-minute isometric effort, one-repetition maximum, relative strength) or subjective (e.g., years training, days per week training) if objective information was not collected.

I think is not clear how the measurements were taken during the isometric efforts. Did the participants stop the isometric effort by dropping the weights to undertake the measurements each 30, 60, 90, and 120 seconds after starting? Did the participants extend their arms at both sides of the body holding the weights? Could a 2-second pause have influenced the maximum effort participants could perform? Consider rewriting this explanation.

I believe it could be interesting for the reader to have reference values for the interpretation of the effect sizes. Discussion on these values can be found in the article of Lakens (2013).

Please, consider including significance (at least with symbols) in Table 1.

Although there is controversy on the effect that the sex of the participants could have on the intraocular pressure, in one study the sex was found to be a significant predictor of the intraocular pressure variations (Gene-Morales et al., 2021). Could these differences appear in the parameters analyzed in the present study? This could be indicated in the future directions section. Also, different muscle adaptations can be obtained with isometric contractions at different joint angles (Oranchuk et al., 2019). It could be interesting to study ocular adaptations to isometric efforts at different joint angles in future works.

Specific comments:
Line 112, the reference [19] is placed after the full stop.

Line 137, a comma is missing between the parenthesis and before as it is a list of three or more words.

Additional references used:
Gene-Morales, J., Gené-Sampedro, A., Martín-Portugués, A., & Bueno-Gimeno, I. (2021). Do age and sex play a role in the intraocular pressure changes after acrobatic gymnastics? Journal of Clinical Medicine, 10(20), 4700. https://doi.org/10.3390/jcm10204700

Lakens D. (2013). Calculating and reporting effect sizes to facilitate cumulative science: a practical primer for t-tests and ANOVAs. Frontiers in psychology, 4, 863. https://doi.org/10.3389/fpsyg.2013.00863

Oranchuk, DJ, Storey, AG, Nelson, AR, Cronin, JB. Isometric training and long-term adaptations: Effects of muscle length, intensity, and intent: A systematic review. Scand J Med Sci Sports. 2019; 29: 484– 503. https://doi.org/10.1111/sms.13375

Reviewer 2 ·

Basic reporting

The manuscript “Immediate and cumulative effects of upper-body isometric exercise on the cornea and anterior segment of the human eye” aimed to analyze the short-term effects of 2-minute upper-body isometric efforts at two different intensities, on corneal and anterior eye morphology. There is undoubtedly a need to examine the effect of isometric exercise on eye function, especially in the context of the importance in prevention and management of ocular diseases (e.g. glaucoma). Thus, I believe your study will be of significant interest to the readers of PeerJ. The manuscript is well written, easy to follow, succinct and adheres to the required format. I would like to congratulate the authors on their exemplary work. In my opinion, it requires only minor revisions, mainly in the discussion section. I hope my comments will be beneficial to the authors.

Comments
1. Is there a possibility that the ages of participants influenced the results? In the discussion section, please consider adding a limitation such as an age range,i.e. in this cross-sectional study only young healthy people participated (convenience samples) and causality cannot be fully inferred.
2. Please consider adding a description of the physiology mechanism for possible long-term consequences of the isometric efforts on the cornea and anterior chamber morphology alterations.

Typos
e.g.
Wylȩgała – should be Wylęgała

Experimental design

Generally the experimental design is well done, sample size and statistics are appropriate and predictions well formulated, and technical details are well described. Authors have sufficiently explained each component of the study procedure, and overall the analytical approach was justified. However, I have one doubt regarding the upper-body isometric exercise protocol. Did participants start the experimental session always, with holding the heaviest load? If yes, is it possible that exercises with a higher load could affect the results of those with a lower one ? Was there a warm-up for the examination session performed?

Validity of the findings

The validity of the findings was enhanced by the use of standardized methods and thoughtful research design. The results are reliable, and importantly, they constitute the starting point for undertaking clinical trials.

---

## Round 0.2 · accepted · Accept

Thanks to the authors for adressing all comments. The article is suitable for publication.

Reviewer 1 ·

Basic reporting

No comment.

Experimental design

No comment.

Validity of the findings

No comment.

Additional comments

All the comments raised by the reviewer have been properly addressed. Thank you.

Reviewer 2 ·

Basic reporting

no comment

Experimental design

no comment

Validity of the findings

no comment

Additional comments

I accept the answers and I have no further comments.